# Effects of Flaxseed and Multi-Carbohydrase Enzymes on the Cecal Microbiota and Liver Inflammation of Laying Hens

**DOI:** 10.3390/ani11030600

**Published:** 2021-02-25

**Authors:** Mazhar Hussain Mangi, Tariq Hussain, Muhammad Suhaib Shahid, Naveed Sabir, Muhammad Saleem Kalhoro, Xiangmei Zhou, Jianmin Yuan

**Affiliations:** 1Key Laboratory of Animal Epidemiology and Zoonosis, Ministry of Agriculture, National Animal Transmissible Spongiform Encephalopathy Laboratory, College of Veterinary Medicine, China Agricultural University, Beijing 100193, China; drmazharmangi114@gmail.com (M.H.M.); drtariq@aup.edu.pk (T.H.); naveedsabir@upr.edu.pk (N.S.); 2College of Veterinary Sciences, The University of Agriculture Peshawar, Peshawar 25130, Pakistan; 3State key Laboratory of Animal Nutrition, College of Animal Science and Technology, China Agricultural University, Beijing 100193, China; suhaib_shahid@yahoo.com; 4Department of Animal Produces Technology, Sindh Agriculture University Tando Jam, Hyderabad 70050, Pakistan; saleemkalhoro@yahoo.com

**Keywords:** microbiota, corn diet, wheat diet, flaxseed, enzyme, liver inflammation

## Abstract

**Simple Summary:**

Wheat and flaxseed are used worldwide to produce omega-3 (ω-3) enriched poultry meat and eggs, however, wheat and flaxseed contain some anti-nutritional factors (ANFs). In addition, the supplementation of feed additive including enzymes usually alleviate the deleterious influence of ANFs. Therefore, we conducted the current study of laying hens fed with two diets (corn/flaxseed and wheat/flaxseed, supplemented with three enzymes), for a period of 10 weeks. Here, we found a clear increase in the fat weight of birds fed with corn diet as compared with wheat diet. Moreover, a high level of secretory IL-1β, IL-6, and IL-10 and comparatively higher inflammatory changes in the liver tissue were found in birds fed with corn diet as compared with wheat diet. The gut microbial composition of hens fed with corn diet was clearly different than that of birds fed a wheat diet. In conclusion, our findings suggest that inflammatory changes in laying birds were mediated by a corn diet with flaxseed and enzymes instead of a wheat diet. Additionally, in the wheat-fed group, enzyme-b and -c showed more encouraging results as compared to enzyme-a indicating that wheat diet might be a preferable diet for commercial layers poultry farms.

**Abstract:**

Background: The use of wheat and flaxseed to produce omega-3 (ω-3) enriched poultry meat and eggs is very popular in the world. However, wheat and flaxseed contain some anti-nutritional factors (ANFs), and enzymes are usually used to alleviate the deleterious influence of ANFs. Method: A 2 × 3 two factors design was used in the experiment. A total of 540 twenty-week-old Nongda-3 laying hens were randomly allocated to six dietary treatments, two diets (corn/flaxseed and wheat/flaxseed), and three enzymes (enzyme-a contains neutral protease 10,000, xylanase 35,000, β-mannanase 1500, β-glucanase 2000, cellulose 500, amylase 100, and pectinase 10,000 (U g^−1^); enzyme-b contains alkaline protease 40,000 and neutral protease 10,000 (U g^−1^); enzyme-c contains alkaline protease 40,000, neutral protease 10,000, and cellulase 4000 (U g^−1^). Results: There was an interaction between dietary treatment and supplemental enzymes for liver weight and liver inflammatory cytokines of broilers. A significant increase was observed in the fat weight of birds fed a corn diet as compared with a wheat diet. A corn diet and wheat diet with the addition of enzyme-a (*p* < 0.001) showed the highest level of liver fat followed by enzyme-c (*p* < 0.01) and enzyme-b. Moreover, a high level of secretory IL-1β, IL-6, and IL-10 and comparatively higher inflammatory changes in the liver tissue were found in birds fed a corn diet as compared with a wheat diet, and enzyme-b showed more beneficial effects as compared with enzyme-a and -c. The gut microbial composition of hens fed a corn diet was significantly different than that of birds fed a wheat diet. *Bacteroides* were significantly (*p* < 0.05) abundant in the corn-fed birds as compared with wheat-fed birds. However, Firmicutes were less abundant in the wheat-fed birds than the corn-fed birds (16.99 vs. 31.80%, respectively). The microbial community at the genus level differed significantly in the dietary groups and we observed that *Bacteroides* are the predominant cecal microbiota. Kyoto Encyclopedia of Genes and Genomes (KEGG) pathways of co-factors, carbohydrates, vitamins, protein, and energy were expressed at slightly higher levels in the microbiota of the wheat-fed birds, whereas, metabolic pathways for nucleotides, lipids, and glycine were expressed at higher levels in the wheat-fed birds. Furthermore, expression of the growth and cellular processes pathway and endocrine system pathway levels were predicted to be higher for the wheat-fed group as compared with the corn-fed group. Conclusions: In conclusion, our findings suggest that inflammatory changes in laying birds were mediated by a corn diet with flaxseed and enzymes instead of a wheat diet. Additionally, in the wheat-fed group, enzyme-b and -c showed more encouraging results as compared to enzyme-a.

## 1. Introduction

The preference of consumers for omega-3 (ω-3) fatty acids has increased in recent years due to several health benefits [1]. Flaxseed is a perfect cradle of ω-3 polyunsaturated fatty acids (PUFAs), especially α-linolenic acid (ALA) [2], which is readily used to produce ω-3 enriched poultry meat and eggs [3,4] and to supply healthier poultry products for human [5]. However, flaxseed is a rich source of nonstarch polysaccharides (NSPs), a water-soluble mucilage [6]. NSPs increase intestinal viscosity and decrease the digestibility of nutrients [7,8]. In addition, flaxseed has been associated with anti-nutritional factors (adhesive, cyanogen glycosides, and trypsin and phytic corrosive inhibitors), which limits its utilization in poultry feed [9]. Similarly, lower omega-6 PUFAs (ω-6PUFAs) in wheat have been used for the base diet to produce ω-3 eggs. There are also as many soluble NSPs as ANFs. A previous study showed that egg production could be significantly reduced in layer flocks when the flaxseed inclusion rate was more than 8% [10]. Flaxseed meal has also been shown to reduce fatty acid digestibility, and stimulated malabsorption of primary bile acids and excretion of secondary bile acids [11] and a flaxseed diet has been shown to cause inflammation by altering the gut microbiota of Peking ducks [12].

The addition of an enzyme blend is helpful to diminish the harmful effects of ANFs, and to enhance the production of laying hens [2]. Previously, protease has been used alone to hydrolyze the trypsin inhibitor of flaxseed. The multi-carbohydrase mixes seem to be more effective at degrading insoluble NSPs in cell walls than individual enzymes [13,14].

Liver inflammation has been detected in commercial laying hens and is frequently the major cause of mortality in healthy flocks, up to 5% during the laying cycle [15]. Adipose tissue plays a significant role in inflammation and immunity due to the production and release of proinflammatory molecules such as interleukin-6 and tumor necrosis factor and anti-inflammatory molecules (e.g., adiponectin and IL-10) from the infiltration of immune cells and adipocytes [16]. An excessive amount of lipid accumulation leads to liver rupture, and death frequently occurs during the increased abdominal pressure of egg laying [17]. To produce ω-3 eggs, 10% flaxseed is often used in the diet, however, the enriched lipid in flaxseed may cause liver inflammation. However, it has been shown that flaxseed’s linolenic acid, i.e., ω-3 fatty acid, has a role in the regulation of immune functions and inflammation [18].

The 16S rDNA sequencing is a useful tool to obtain complete data of bacterial communities and is necessary for gaining new insights into the biological and ecological roles of microbiota [19]. However, a comparative investigation of microbiota profiles with their biochemical properties and regulation of inflammatory markers under adequate controls would produce reliable data. Taking into consideration the previous studies, the current study was designed to investigate the effect of three different enzymes and 10% flaxseed mixed with corn or wheat on the cecal microbiota and liver inflammation of laying hens.

## 2. Materials and Methods

### 2.1. Ethics Statement

All the animal trials were carried out under the protocol of the Chinese Regulations of Laboratory Animals. The Laboratory Animal Ethical Committee approved the experimental animal protocols (AW04110202-1) of the China Agricultural University.

### 2.2. Experimental Feed and Enzymes

The two main diets were based on corn and wheat; each diet was supplemented with 10% whole flaxseed. The corn and wheat diets were supplemented with three different multi-carbohydrase enzymes (200 g/t). Treatments A, B, and C were based on corn and 10% flaxseed, and supplemental enzyme-a, enzyme-b, and enzyme-c, respectively. Treatments D, E, and F were based on wheat and 10% flaxseed, and supplemental enzyme-a, enzyme-b, and enzyme-c, respectively. The diets were formulated to be isocaloric and iso-nitrogenous and to meet or exceed [20] requirements for laying hens (Table 1). All the diets were fed to the hens for 10 weeks.

Enzyme-a contained neutral protease 10,000, xylanase 35,000, β-mannanase 1500, β-glucanase 2000, cellulose 500, amylase 100, and pectinase 10,000 (U g^−1^). Enzyme-b contained alkaline protease 40,000 and neutral protease 10,000 (U g^−1^). Enzyme-c contained alkaline protease 40,000, neutral protease 10,000, and cellulase 4000 (U g^−1^).

### 2.3. Birds and Experimental Design

A total of 540 20-week-old Nongda-3 laying hens were used in a 10-week feeding assay. Six replicate cages of 15 hens were randomly assigned to one of 6 dietary treatments. Feed and water were provided to all experimental birds, ad libitum, 16-h light, and 8-h dark periods as a standard schedule for laying birds. The duration of the experiment was 10 weeks.

### 2.4. Sample Collection Histological and Serum Analysis

Six birds from each replicate were randomly selected for sampling. Blood samples were aseptically collected from the wing veins of birds, serum was extracted by centrifugation and stored at −80 °C for further analysis of cytokines. Birds were sacrificed by cervical dislocation and liver and cecal samples were collected aseptically. For the histological examination, the tissues were fixed in 10% formaldehyde solution, embedded in paraffin, and cut into sections (5 µm) using a microtome [21]. Thin sections of liver tissues were mounted on glass slides, deparaffinized, and stained with the Hematoxylin and Eosin (H&E). Finally, H&E-stained sections were visualized under low and high power of microscope by using an Olympus DP72 microscope fitted with a camera. The percent of area occupied by various inflammatory cells in the liver tissue of birds fed various diet were calculated by using Image J software. Commercially available chicken enzyme-linked immunosorbent assay kits (CUSABIO Company, Beijing, China) were used to measure the concentrations of IL-1β, IL-6, and IL-10 in the blood serum samples. The concentration of cytokines was measured by using a standard curve that was obtained by using 2-fold dilutions of the standard for each independent experiment.

### 2.5. DNA Extraction

A QIA amp TM Fast DNA Stool Mini Kit (QIAGEN, Hilden, Germany) was used to extract DNA from 180–220 mg of each sample, according to the manufacturer’s instructions. The 16S rDNA gene amplicons were performed by the Realbio Genomics Institute (Shanghai, China) using an Illumina Hi Seq PE250 (San Diego, CA, USA) platform. The V3–V4 region of the 16S rDNA gene was amplified using the universal primers 34 1F (CCTACGGGRSGCAGCAG) and 806R (GGACTACVVGGGTATCTAATC).

### 2.6. Statistical Analysis

Data were analyzed by two-way ANOVA using the Graph Pade Prism 5.0 [22] and Statistix 8.1 software (Tallahassee, FL, USA) and two-way analyses of variance (ANOVA) shadowed by the Tukey multiple comparison. Results were considered significantly different at *p* < 0.05 between the trail groups. Using QIIME software, called the UCLUST sequence alignment tool [23], previously obtained sequences were merged with 97% sequence similarity and an operational taxonomic unit (out), and the highest abundance sequence in each OTU as the representative OUT sequence was selected. Then, based on the number of sequences, each sample has an OTU, a matrix file (i.e., OTU table) of OTU abundance in each sample was constructed, and the matrix file was converted into a “BIOM” (biological observation matrix), a file format that is easier to transfer, store, and compatible with the other analysis tools.

## 3. Results

### 3.1. Effect of Corn and Wheat Diet Supplemented with Flaxseed and Enzyme on the Liver and Fat Weight of Laying Hens

There was a significant effect of diet and enzymes on liver and fat weight, as shown in Figure 1. There was an interaction between dietary treatment and supplemental enzymes for liver weight of broilers. The liver weight of the corn-fed groups were significantly higher than those of wheat-fed diet groups (Figure 1a). The liver weight of the corn-fed group with enzyme-c were significantly higher (*p* < 0.01) than those of the wheat-fed groups, followed by corn-fed groups with enzyme-a and enzyme-b were also higher (*p* < 0.05) as compared with the wheat-fed groups.

In addition, the corn- and wheat-fed groups with the addition of enzyme-a (*p* < 0.001) showed the highest level of liver fat followed by enzyme-c (*p* < 0.01) and enzyme-b (*p* < 0.05, Figure 1b).

### 3.2. Effect of Corn and Wheat Diet Supplemented with Flaxseed and Enzyme on Inflammatory Cytokines in Laying Birds

There was an interaction between diet treatment and supplemental enzymes for inflammatory cytokines in laying birds (Figure 2). The IL-1β (Figure 2a), IL-6 (Figure 2b), and IL-10 (Figure 2c) levels in birds fed a corn diet were significantly higher than the levels in birds fed a wheat diet. In addition, the corn diet supplemented with enzyme-c (*p* < 0.001) showed the highest level of IL-1β followed by enzyme-a (*p* < 0.01) and enzyme-b (*p* < 0.05, Figure 2d). In addition, enzyme-a and -b in the corn diet led to the highest expression (*p* < 0.001) of IL-6 as compared with the corn diet supplemented with enzyme-c (*p* < 0.05, Figure 2f). The level of IL-10 in the birds fed the corn diet containing enzyme-c was significantly higher (*p* < 0.001) than the levels in the birds fed the corn diets supplemented with enzyme-a and enzyme-b (*p* < 0.01, Figure 2h). However, the birds fed the wheat diets supplemented with enzyme-a and enzyme-b showed a higher level of IL-1β than the birds fed the wheat diet supplemented with enzyme-c (*p* < 0.05). Birds fed the wheat diet supplemented with enzyme-b had the lowest level of IL-6 as compared with the birds fed the wheat diets supplemented with enzyme c, followed by enzyme-a. The level of IL-10 in the birds fed the wheat diet supplemented with enzyme-c was significantly lower than the birds fed the wheat diets supplemented with enzyme-b, and enzyme-a (*p* < 0.05).

### 3.3. Histopathological Analyses of Liver Tissues of Birds Fed Corn and Wheat Diets Supplemented with Flaxseed and Enzymes

There was variation in the infiltration of various inflammatory cells in the liver sections of birds fed with different diets supplemented with flaxseed and enzymes (Figure 3). In particular, all corn-fed birds showed increased inflammatory signs in the H&E sections of liver tissues as compared with wheat-fed birds (Figure 3b). As shown in Figure 3b, the corn diet with enzyme-a significantly damaged the maximum area of the liver tissue (30.9%) as compared with all other diets. In contrast, there were fewer inflammatory lesions in the liver tissues of wheat-fed birds with all three enzymes (Figure 3b). These results suggest that a corn diet supplemented with flaxseed and enzymes, mediates inflammatory changes in the liver of laying birds, however, enzyme-b showed more beneficial effects as compared with enzyme-a and -c (Figure 3d). In the wheat-fed groups with enzymes (a, b and c), there was more benefit for the groups with enzyme-b and -c groups than the enzyme-a group. Collectively, these results suggested that enzyme-b and -c were beneficial in the wheat diet as compared with that in the corn diet.

### 3.4. Effect of Corn and Wheat Diet Supplemented with Flaxseed and Enzyme on OTU Topographical Difference in 16S rDNA Sequence Richness in Laying Birds

A total of 36 ceca samples, six from each dietary group were collected for analyses of the microbiota. A total of 1,318,942 sequence reads with a median length of 36,637 nt was observed from all cecal samples. The sequences were further clustered into 786 OTUs using a 97% similarity cutoff. The clustering analyses of 31 OTUs with the highest default abundances revealed similarities as well as differences between the samples. In our study, differences in species richness between the groups were identified by using an abundance-based coverage estimator, the Chao1, Simpson’s index (D), and Shannon’s index (H) (Table 2). The Simpson index of the group fed a corn diet was observed to be significantly higher (*p* < 0.05) than the group fed a wheat diet. The results showed that species diversity was comparatively more abundant in birds fed a corn diet than birds fed a wheat diet. The rarefaction curves of the microbiota of 36 samples were sufficiently large to estimate phenotypic richness and microbial community diversity at a similarity threshold of 97% (Figure 4). The rarefaction curves showed that the sampling effort had sufficient sequence coverage to accurately describe the bacterial composition of each group (Figure 4).

Gut microbiota from the cecal samples of corn and wheat-fed groups was assessed by 16S rRNA gene sequencing technique. ACE, Chao1, and Shannon index were used to assess the alpha diversity of the cecal microbiota in the corn-fed and wheat-fed groups with different enzymes and 10% Flaxseed. No significant difference in the biodiversity of the microbiota was found between the corn-fed and wheat-fed groups (Table 2).

### 3.5. Effect of Corn and Wheat Diet Supplemented with Flaxseed and Enzyme on Microbial Beta Abundance-Based Coverage Estimator Analyses in Laying Birds

The sequence-based rarefaction curves based on the phylogenetic diversity metric confirmed a significant difference in the diversity of the corn-fed and wheat-fed groups (Wilcoxon rank-sum test) (Figure 5). The distribution of corn-fed groups A, B, and C were separated from the wheat-fed groups D and F, while there was no difference in wheat-fed group E.

### 3.6. Effect of Corn and Wheat Diet Supplemented with Flaxseed and Enzyme on Microbial Beta Diversity Analyses in Laying Birds

Beta diversity analyses of the microbial community in all cecal samples are shown in PCA plots, indicating 30.25% PC1 and 11.73% PC2 (Figure 6). The PCA results presented that the gut microbiota structure had changed, and most of the corn-fed group samples were concentrated in the third and fourth quadrant, while the wheat-fed groups were concentrated in the first and the second quadrants. The CR group was primarily concentrated in the first quadrant, and the remaining groups were concentrated in the second and the third quadrants. According to the PCA results, the corn-fed groups A, B, and C were separated from wheat-fed groups D, E, and F, whereas the corn-fed groups (A, B, and C), and wheat-fed groups (D, E, and F) were related to each other. Few samples of groups D and E were closely aligned with that of the corn diet groups. These results suggest that the abundance of bacterial species did not differ significantly among groups, however, there was a difference in species diversity, and the gut microbiota structure was altered in corn-fed and wheat-fed birds.

Gut microbiota analysis for relative abundance at the phylum level revealed that the gut microbiota in six dietary groups was dominated by members of Bacteroidetes (45–82%) and Firmicutes (34–50%), followed by Actinobacteria and Proteobacteria in considerably lower abundances (Figure 7a). At the genus level, *Bacteroides*, *Lactobacillus, Provetella, Facillibacterium, and Ruminococcacus* were found to be the prominent genus (Figure 7b).

### 3.7. Taxonomic Composition

Next, we assessed the differences in sequence between the three corn-fed and three wheat-fed dietary groups at the phylum as well as genus level. The c ratios in the corn-fed (A, B and C) groups were 0.5, 1.0, and 0.4 respectively. In addition, the wheat-fed (D, E and F) groups showed 0.3, 0.4, and 0.5, ratio for +10% flaxseed, respectively (Figure 8a). The relative abundance of the genus 5-7N15 was decreased significantly by the wheat diets (*p* < 0.05, FDR ¼ 0.060) (Figure 9a). The abundance of anti-inflammatory bacteria Akkermansia/Allobaculum (*p* < 0.05, FDR < 0.05) was increased in the corn-fed groups A (supplemented with enzyme-b) and B (supplemented with enzyme-b) but it was decreased in the corn-fed groups C (supplemented with enzyme-c) (Figure 9b). The allobaculum ratio in the corn-fed (A, B, and C) and wheat-fed (D, E and F) groups were 7, 2, and 0 and 0, 0, and 0, respectively, (Figure 9c). In contrast, the relative abundance of proinflammatory bacteria Bacteroides (*p* ¼ 0.0582, FDR ¼ 0.1020) increased in all groups (Figure 9e). In addition, the abundances of anti-inflammatory bacteria Bifidobacterium (*p* < 0.05, FDR < 0.05), Campylobacter (*p* < 0.02, FDR < 0.05), Cetobacterium (*p* ¼ 0.0681, FDR ¼ 0.1204), Clostridium (*p* < 0.02, FDR < 0.03), Coprobacillus (*p* < 0.01, FDR < 0.02), Coprococcus (*p* < 0.05, FDR < 0.05), Deulfovibria (*p* < 0.05, FDR < 0.05), Dorea(*p* < 0.02, FDR < 0.03), Elusimicrobium (*p* < 0.01, FDR < 0.02), Faecalibacterium (*p* < 0.05, FDR < 0.05), Oscillospira (*p* < 0.03, FDR < 0.05), PFNzo (*p* < 0.02, FDR < 0.02), Rikenella (*p* < 0.05, FDR < 0.05), and Slackia (*p* < 0.02, FDR < 0.02) increased in all groups of corn-fed and wheat-fed birds. However, the relative abundance of proinflammatory bacteria, Prevotella (*p* < 0.02, FDR < 0.04), Coprobacillus (*p* < 0.02, FDR < 0.05), and Peptococcus (*p* < 0.05, FDR < 0.05) showed an increase in corn-fed groups (B and C) as compared with all wheat-fed groups. At the genus level, 55 genera differed significantly among the groups. The genus Actinobacteria was dominant in the wheat-fed group, accounting for 49.59% of the observed genera, whereas Bacteroidetes, Deferribacteres, and Fusobacteria were the observed genera in the corn-fed groups (Figure 8). Proinflammatory genera including Proteobacteria (*p* ¼ 0.0348, FDR ¼ 0.0705), Spirochaetes (*p* ¼ 0.0572, FDR ¼ 0.1015), Synergistetes (*p* < 0.01, FDR < 0.05), and TM7 (*p* < 0.05, FDR < 0.101.5) were enriched in the corn-fed groups.

### 3.8. Effect of Corn and Wheat Diet Supplemented with Flaxseed and Enzyme on Microbiota Community Composition in the Cecum in Laying Birds

Cecal samples were analyzed for the community composition of the microbiota. Therefore, R software was used to assess various genera. Initially, the microbial community was clustered according to the similarity basis, and then arranged horizontally through clustering results. Then, the classification units were also clustered considering the similarity of each other in different samples and arranged vertically in the clustered plot. As shown in Figure 10, the red color area in the plot represents the genus with higher abundance in the corresponding sample, while the green color indicates the genus level with lower abundance. The culture plot showed that populations of *Lactobacillus* and Firmicutes increased remarkably between groups of both diets, respectively. Moreover, the Bacteroidetes were more abundant in the birds fed a corn diet than the birds fed a wheat diet.

### 3.9. Comparison of the Cecal Microbiota KEGG Pathways of the Corn-Fed and Wheat-Fed Groups

The functional profile databases of KEGG PATHWAYS were used for predicting the metabolic pathways of microbiota in the corn-fed groups and wheat-fed groups, focusing only on the six KEGG pathways of metabolism, environmental processing, genetic processing, organismal systems, cellular processes, and human diseases. According to the predictions of (PICRUSt), the annotation information for each corresponding functional spectrum database of each group is presented (Figure 11). Metabolic pathways of co-factors, carbohydrates, vitamins, protein, and energy were predicted to be expressed at slightly higher levels in the microbiotas of the wheat-fed groups; metabolic pathways for nucleotides, lipids and glycine were predicted to be at higher levels in the wheat-fed group. Whereas, levels of expression of the growth and cellular processes pathways, as well as the endocrine system pathway, were predicted to be higher for the wheat-fed groups than those for the corn-fed groups (Figure 11b,c). Moreover, levels of expression of the cell motility were predicted to be higher in the corn-fed groups than those for the wheat-fed groups (Figure 11b).

## 4. Discussion

It is well known that the basic ingredients of poultry diets contain seeds from various plants. In the present study, we investigated the efficacy of dietary flaxseed with different multi-carbohydrase enzymes in corn and wheat diets on the induction of inflammatory cytokines, inflammation in liver, and its association with gut microbiota in laying hens. Early studies have reported that flaxseed could be added to poultry diets to enrich their products for human consumption [24]. The NSP compounds of the flaxseed cannot be digested by the chicken due to a lack of NSP hydrolyzing enzymes. The NSP degrading enzymes break the NSP of the plant cell wall and release the entrapped nutrients into the gut. In addition to NSPs, the presence of ANFs in the flaxseed worsens the performances of birds [25]. Previously, the use of a corn diet which is high in calories can result in obese chicken, fatty liver, and cause liver inflammation [26]. Similarly, we found that corn-fed birds as compared with wheat-fed birds showed high-fat contents, increased liver fat, and inflammation in the liver with higher levels of inflammatory cytokines, indicating better dietary effects of a wheat diet as compared with a corn diet. Multiple enzymes which contain cellulase, xylanase, and β-glucanase carbohydrase in nature have been used in the poultry industry to alleviate the anti-nutritional factors and improve the utilization of dietary energy and protein, leading to enhanced performance, cecal microflora and intestinal viscosity, digestibility, and resulting in greater availability of nutrients in the feed [27]. These findings suggest that these interventions exhibit preventive and therapeutic potential. Several studies have emphasized the diversity and composition of cecal microbiota in poultry birds that might be associated with the modulation of inflammatory reactions in response to various ingredients of poultry diets. Our results are in line with reports from a previous study that birds fed with a diet containing flaxseed had decreased gut microbial alpha diversity in laying hens [28]. In modern times, the potential applications of flaxseed, enzymes in aquatic animals, and rat models have been increasingly explored. In our study, we examined the effect of a corn and a wheat diet supplemented with 10% flaxseed and enzymes (a, b, and c) on the diversity of cecal microbiota and inflammatory changes in the liver tissue of laying birds.

We found that a corn diet supplemented with flaxseed and enzyme promoted Gram-positive bacteria and *Bacteroides* at the genus level. However, a significant difference was observed in Lactobacillus, Oscillospisa, Firmicutes, and Actinobacteria, in both corn-fed and wheat-fed laying birds. The attenuation in major colonizers of the small intestine could also be associated with consequences in nutrition at the onset of egg laying. Interestingly, except for a switch in the dominance of two main genera, microbiota in the small intestine remained rather constant and did not develop with chicken age. This was different from the microbiota in the cecum, where it developed with increasing age [29,30,31]. It has been studied that Firmicutes were the first family of the phylum Bacteroidetes which appeared in the ceca, later replaced with representatives of families Bacteroideaceae and Actinobacteria. This seems to be a common developmental profile, since high Romboutsia abundance was recorded in several studies in 3- to 5-week-old chickens [32,33].

None of the monitored diets displayed clinical signs of any infection. However, despite the absence of clinical signs, the adult hens showed suboptimal performance. In the current study, the microbiota of ceca did not exhibit any extensive differences in chickens fed with both experimental diets, but *Bacteroides* dominated the cecal microbiota. Until now, there has been little information available about *Bacteroides* infections in chickens. Furthermore, the classification of genus *Bacteroides* showed that *Peptococcus* was the species of increased abundance in the ceca of chickens in birds fed a corn diet. Moreover, the interactions of Bacteroides and particular hosts are dependent not only on species but even on exceptional clones within species [34]. In contrast, Prevotella has been detected in wild birds at a relatively low incidence [19,35]. A previous study also considered the zoonotic potential of *Peptococcus* [36]. There is a report of mild lesions in the caeca of sacrificed chickens infected with Helicobacter [37]. In adult hens, the microbiota differed from that of chickens before reaching sexual maturity. The Lactobacillus level was greater than the Prevotella level in the ceca, similar to previous reports [13]. In a previous study, Bifidobacterium, Lactobacillus, and Firmicutes were enriched in the flaxseed-fed groups [38]. However, the dominance (around 20% of ceca microbiota) of Firmicutes in the cecum of hens was quite unpredicted, as Firmicutes were not commonly found among chicken microbiota members. Actinobacteria have been previously associated with multifactorial diseases. Firmicutes mortiferous has been isolated from a human patient with multifactorial sepsis [39] and mixed anaerobic infection of the thyroid gland [40]. In developing countries, campylobacter infection is an important cause of poultry and human bacterial enteritis. In the present study, the average ratios of Firmicutes to Bacteroidetes in the corn-fed groups were higher than those observed in the wheat-fed groups. Previous reports also observed an increased Firmicutes/Bacteroidetes ratio related to high-fat diets and obesity [41] and this increased ratio has been closely linked to *clostridium difficile*-associated diarrhea [42]. Proinflammatory bacteria, including a member of the taxa Cetobacteriam, Peptococcus, and Prevotella were also enriched in the corn diet groups. Members of the class Negativicutes possess outer membranes with lipopolysaccharides that are associated with periodontitis [43]. The family Veillonellaceae is positively correlated with inflammation [44]. Though mostly appearing as a commensal, if present in immune-compromised individuals, Firmicutes mortiferous may spread beyond the intestinal tract and induce sepsis and inflammatory response in its host. On the one hand, the activation of neutrophils and Kupffer cells stimulate the production of inflammatory cytokines and oxidative stress such as IL-1β, IL-6, and IL-10 that cause apoptosis and necrosis of hepatocytes [45,46]. On the other hand, our research showed a significant increase in the levels of IL-1β and IL-6 in the blood serum of birds fed a corn diet supplemented with flaxseed and enzymes, demonstrating that dietary flaxseed alleviated hepatic inflammation level of Il-1β and Il-6. This result validates that a corn diet supplemented with 10% flaxseed and enzymes (a, b, and c) induced aggravated inflammatory response in laying hens as compared with a wheat diet supplemented with 10% flaxseed and enzymes (a, b, and c). Furthermore, we investigated the improvement of ceca microbiota in layer birds. The present study revealed that there was a wide range for the composition of microbiota, as each diet developed slightly different microbiota. Thus, we identified two bacterial species as possibly novel opportunistic pathogens of chickens, however, their real significance for poultry gut health will have to be determined by experimental infections. Moreover, next-generation sequencing should be used in cases with non-specific symptoms such as suboptimal body weight and a decrease in egg production.

## 5. Conclusions

These findings indicate that a wheat diet supplemented with 10% flaxseed and enzymes (a, b, and c) is beneficial for the laying hens as compared with a corn diet supplemented with 10% flaxseed and enzymes. Furthermore, these results revealed that this effective treatment was associate with altered gut microbiota and also caused a reduction in liver inflammation. These outcomes also suggest that the application of a wheat diet supplemented with flaxseed and enzyme in the poultry industry would be beneficial for the health of the laying hens, and that enzyme-b and -c show more encouraging results as compared with enzyme-a.

## Figures and Tables

**Figure 1 animals-11-00600-f001:**
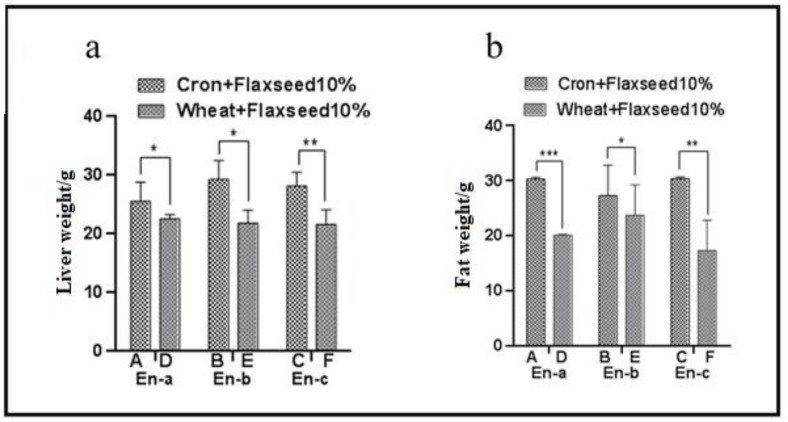
Effect of corn and wheat diet supplemented with flaxseed and enzyme on liver and fat weight in laying hens. Corn and wheat diets were formulated with 10% flaxseed and indicated as Treatments A, B, C and D, E, F, respectively; The two diets were treated, respectively, with three different multi-carbohydrase enzymes, and indicated as enzyme-a, enzyme-b, and enzyme-c. (**a**) liver and (**b**) fat weight of laying hens were calculated after feeding with corn and wheat diets supplemented with enzymes for 28 days Data are expressed as mean ± SEM. * *p* < 0.05, ** *p* < 0.01, and *** *p* < 0.001.

**Figure 2 animals-11-00600-f002:**
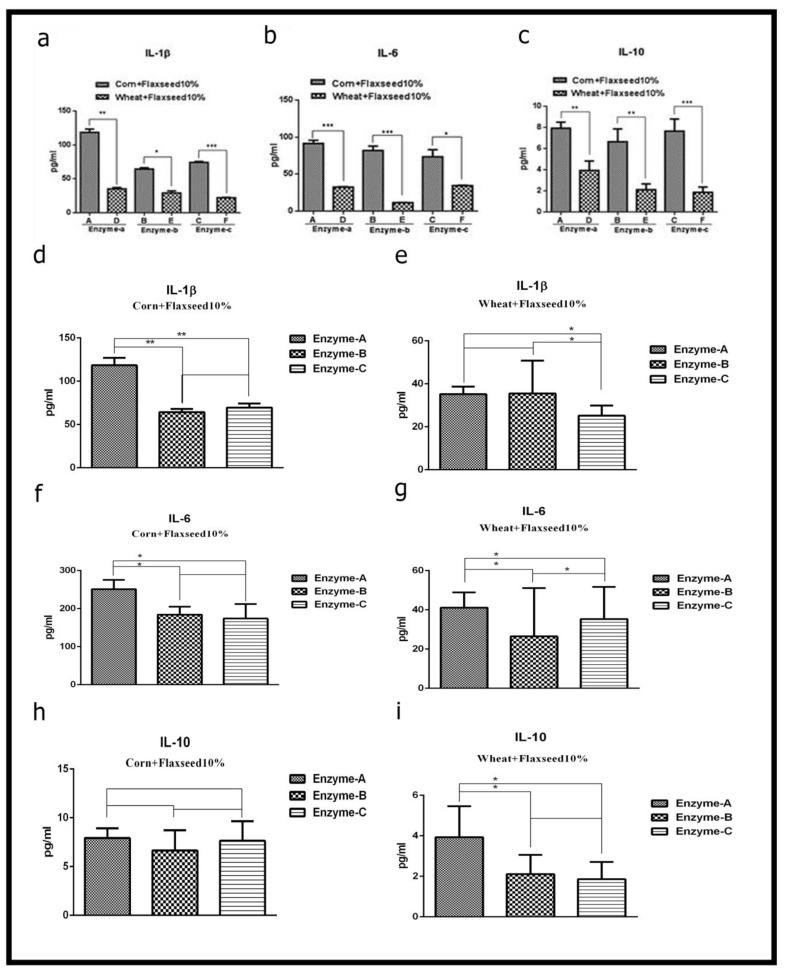
Effect of flaxseed and enzyme supplementation of corn and wheat diet on the induction of cytokines in laying birds. Blood serum samples were collected from laying birds at day 70 after feeding with different diets (*n* = 6). The levels of (**a**) IL-1β; (**b**) IL-6; (**c**) IL-10 were assessed by using ELISA. The level of cytokines in corn diets and wheat diets IL-1β, (**d**,**e**); IL-6 (**f**,**g**); and IL-10 (**h**,**i**) with various treatment enzymes (a, b and c). Data are expressed as mean ± SEM.* *p* < 0.05, ** *p* < 0.01, *** *p* < 0.001, and non-significantly.

**Figure 3 animals-11-00600-f003:**
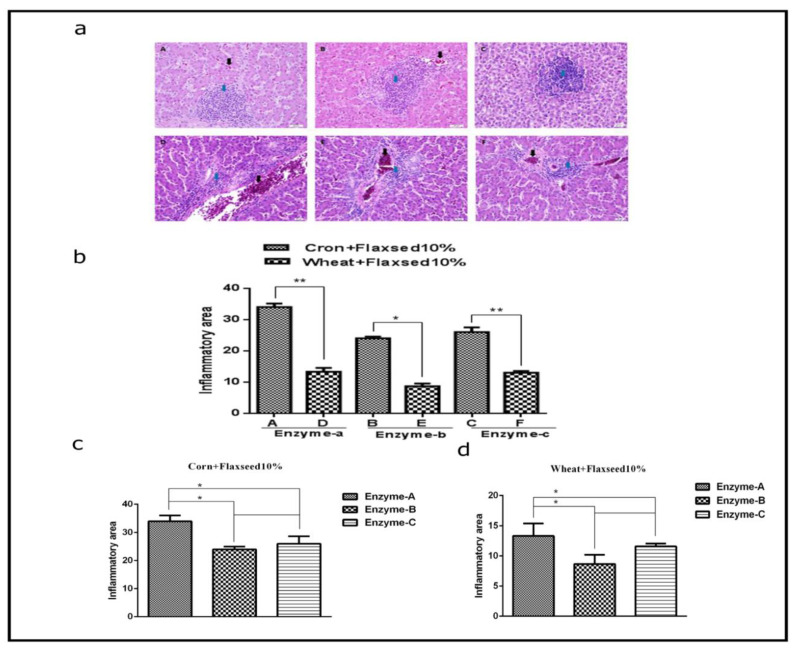
The effects of corn and wheat diet supplemented with 10% flaxseed and enzyme on the histopathology of liver tissues of lying birds. Corn and wheat diets were formulated with 10% flaxseed and 3 different multi-carbohydrase enzymes, as indicated in Table 1. (**a**) Representative images of H&E stained liver sections of various diet-fed birds. Inflammatory cells are marked with blue arrows, while black arrows show the accumulation of RBCs in the liver tissue. (**b**) Percent area occupied by various inflammatory cells in the liver tissue of various diet-fed bird groups. Birds fed the corn diet + 10% flaxseed supplemented with (A) enzyme-a, (B) enzyme-b and (C) enzyme-c (*n* = 6). Birds fed with wheat diet + 10% flaxseed supplemented with (D) enzyme-a, (E) enzyme-b and (F) enzyme-c (*n* = 6); Image J software was used for the analyses of % inflammatory area in the liver tissue of birds fed with (**c**) corn diet + 10% flaxseed and (**d**) wheat diet + 10% flaxseed. Data are expressed as mean ± SD, scale bar 20 µm. Data is expressed as mean ± SEM. (**c**,**d**) * *p* < 0.01, ** *p* < 0.001.

**Figure 4 animals-11-00600-f004:**
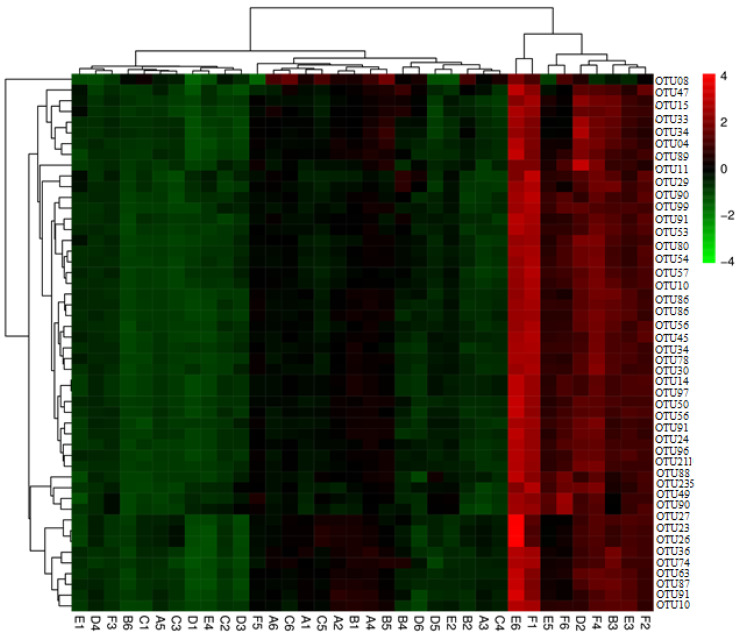
Clustering analyses of cecal samples. Birds were fed with two main diets, corn diet supplemented with (A) enzyme-a, (B) enzyme-b and (C) enzyme-c and wheat diet supplemented with (D) enzyme-a, (E) enzyme-b, and (F) enzyme-c and 10% flaxseed. Cecal samples were collected at day 70 of feeding for analyses of microbiota by using 16S rDNA sequence richness.

**Figure 5 animals-11-00600-f005:**
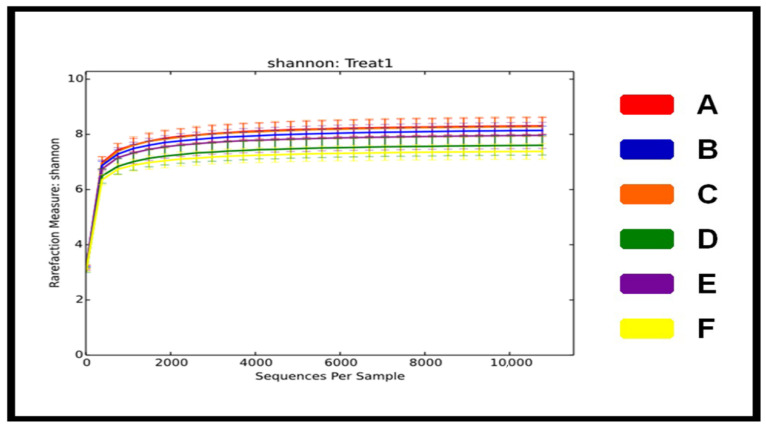
Rarefaction curve sequences indicating the complexities of the microbial communities in the cecal samples of laying birds. Corn diet supplemented with (A) enzyme-a, (B) enzyme-b and (C) enzyme-c and wheat diet supplemented with (D) enzyme-a, (E) enzyme-b, and (F) enzyme-c.

**Figure 6 animals-11-00600-f006:**
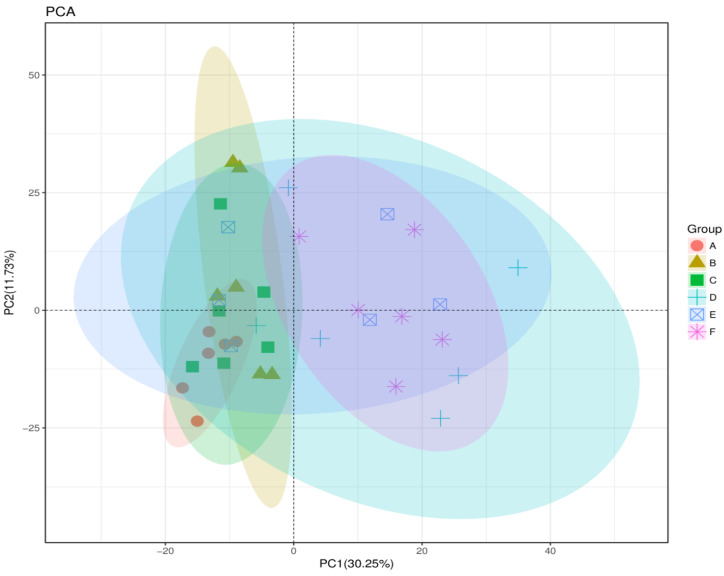
Microbiota composition among different groups at the genus level. PCA analyses by R software test among the corn-fed groups and wheat-fed groups. The degree of clustering reflects the similarity of the groups. The cecal microbiotas of the six groups could be divided into clusters according to community composition.

**Figure 7 animals-11-00600-f007:**
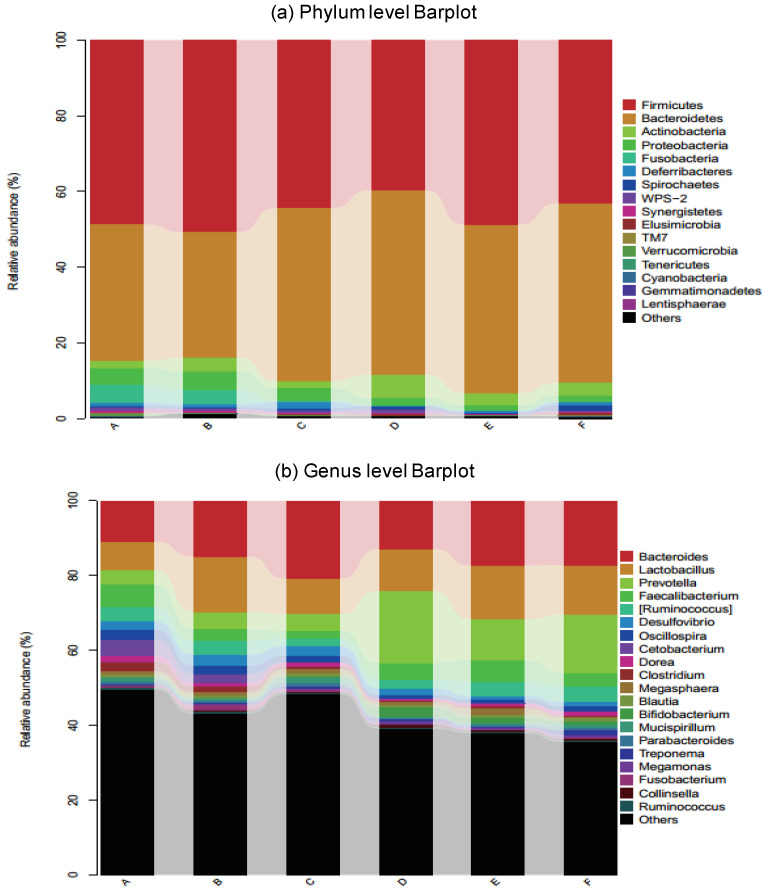
Relative abundance of the dominant bacteria communities in the corn-fed groups (A, B, and C) and the wheat-fed groups (D, E, and F). Each bar chart represents the relative abundance of each group. Each color represents a specific bacteria phylum (**a**) and a specific bacteria genus (**b**).

**Figure 8 animals-11-00600-f008:**
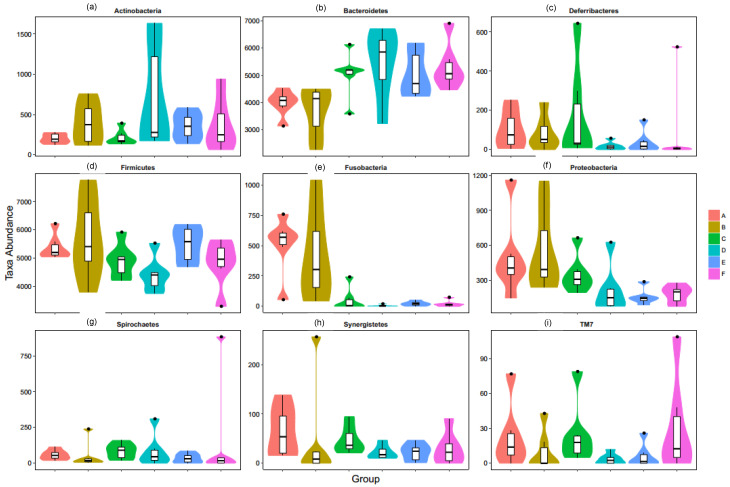
Relative abundance of phylum. (**a**) Actinobacteria; (**b**) Bacteroidetes; (**c**) Deferribacteres; (**d**) Firmicutes; (**e**) Fusobacteria; (**f**) Proteobacteria; (**g**) Spirochaetes; (**h**) Synergistetes; (**i**) TM7. Data are shown as the mean ± SEM. Each color represents the relative abundance of each diet group (A, B, C, D, E and F).

**Figure 9 animals-11-00600-f009:**
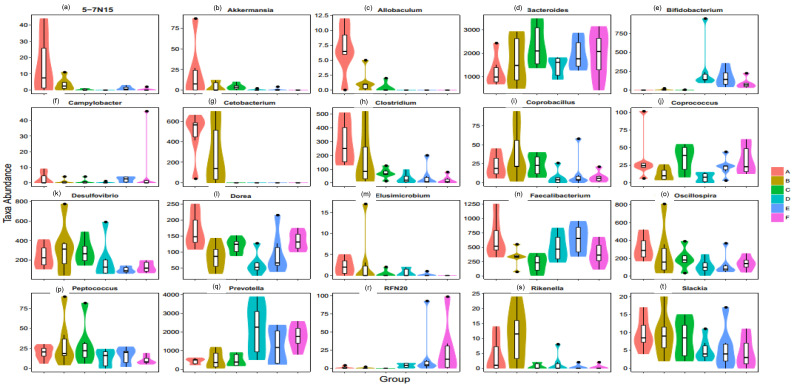
Relative abundances of (**a**) 5-7N15; (**b**) Akkermansia; (**c**) Allobaculum; (**d**) Bacteroides; (**e**) Bifidobacterium; (**f**) Camplylobacter; (**g**) Cetobaccccterium; (**h**) Clostridum; (**i**) Coprobacillus; (**j**) Coprococcus; (**k**) Desulfovibrio; (**l**) Dorea; (**m**) Elusimicrobium; (**n**) Faecalibacterium; (**o**) Oscillospira; (**p**) Peptococcus; (**q**) Prevotella; (**r**) RFNzo; (**s**) Riskenella; (**t**) Slackia. Data are shown as the means ± SEM. Each color represents the relative abundance of each diet group (A, B, C, D, E, F).

**Figure 10 animals-11-00600-f010:**
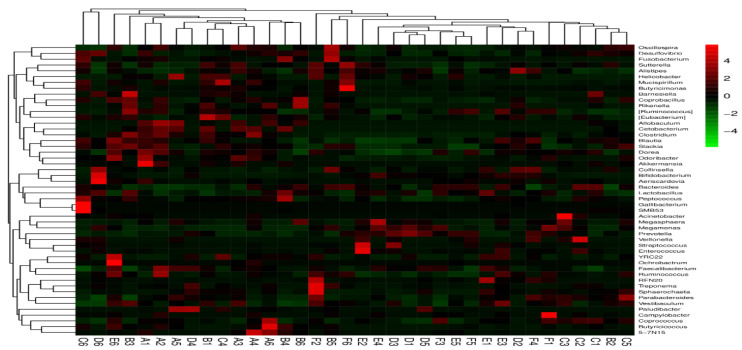
Heat map analyses of cecal samples and the heat map showing the abundances of the top 30 genera were clustered and plotted using R software. Green color represents genera with higher abundances in the corresponding sample, and red color represents genera with lower abundances.

**Figure 11 animals-11-00600-f011:**
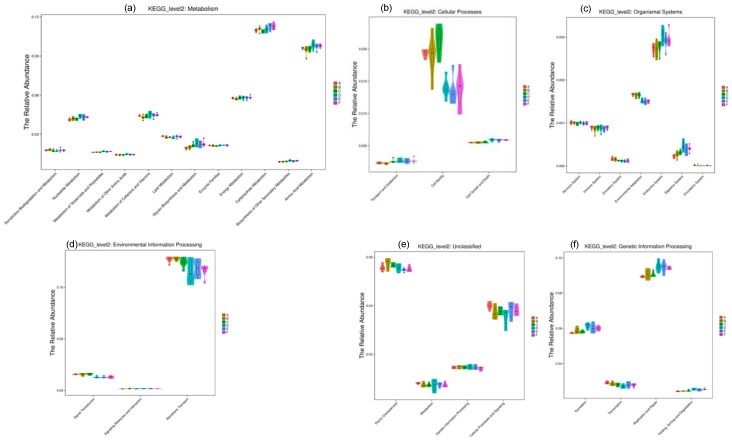
Phylogenetic investigation of communities by reconstruction of unobserved states (PICRUSt) predicted Kyoto encyclopedia of Genes and Genomes (KEGG) at the second level distribution map between the high activities of corn-fed groups and low activity of wheat-fed groups which reflects the number of samples corresponding to the abundance. The predicted KEGG level of (**a**) Metabolism; (**b**) Cellular processes (**c**) Organismal systems (**d**) Environmental information Processing (**e**) Unclassified and (**f**) Genetic information processing are presented for corn-fed and wheat-fed groups.

**Table 1 animals-11-00600-t001:** Ingredients composition and nutrient contents of laying hen diets.

Ingredient	Corn Diet	Wheat Diet
Ingredient basis (%)		
Corn	52	0
Wheat	0	65.2
Soybean Meal	27.00	13.6
Flaxseed	10.00	10.00
Limestone	9.00	9.00
Calcium hydro-phosphate	1.10	0.96
Salt	0.35	0.35
DL-Methionine (99%)	0.17	0.18
L-Lysine sulfate	0.00	0.33
Mineral premix ^1^	0.20	0.20
Vitamin premix ^2^	0.02	0.02
Choline chloride (60%)	0.10	0.10
Selenium yeast	0.02	0.02
Phytase	0.02	0.02
Antioxidant	0.02	0.02
Total	100.00	100.00
Nutrient analyses		
ME, Mkcal/kg	2.758	2.758
Crude protein, %	17.50	17.50
Ca, %	3.89	3.89
Non-phytic acid phosphorus, %	0.26	0.26
Lysine, %	1.04	1.04
Methionine, %	0.53	0.53
M + C, %	0.79	0.79
Threonine,%	0.68	0.68
Tryptophan, %	0.23	0.23
Linoleic acid, %	6.91	6.91

^1^ The mineral premix provided per kg of diet were I, 0.35%; Mn, 100 mg; Fe, 80 mg; Zn, 60 mg; and Cu, 8 mg. ^2^ The vitamin premix provided per kg of diet were vitamin A, 10,000 IU; vitamin D3, 2400 IU; vitamin E, 20 IU; pantothenic acid, 10.00 mg; riboflavin, 6.40 mg; pyridoxine, 3.00 mg; nicotinic acid, 30.00 mg; thiamin, 2.00 mg; vitamin K3, 2.00 mg; folic acid, 1.00 mg; VB12, 0.02 mg; biotin, 0.10 mg.

**Table 2 animals-11-00600-t002:** Effect of corn and wheat diet supplemented with flaxseed and enzymes on alpha diversity estimation of the 16S gene libraries of cecal samples from the 16S sequences in laying birds.

Group	ACE	Chao1	Shannon
A	8908.06 ± 59.983	5.91 ± 2.65	8995.45 ± 157.24
B	9870.11 ± 103.29	5.89 ± 1.86	10,140.47 ± 102.62
C	8988.74 ± 147.44	5.90 ± 2.85	9038.32 ± 151.09
D	8608.61 ± 75.148	5.81 ± 5.71	8991.76 ± 77.108
E	9828.4 ± 182.75	5.88 ± 3.90	10,389.73 ± 209.59
F	6999.98 ± 56.034	5.83 ± 2.64	7355.83 ± 59.983

Note: The rows of the corn diet and wheat diet groups are the average alpha diversity value of each group. The *p*-value is determined from the Kruskal–Wallis test.

## Data Availability

Data sharing not applicable.

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
