# Peer review of "Effects of Flaxseed and Multi-Carbohydrase Enzymes on the Cecal Microbiota and Liver Inflammation of Laying Hens"

_animals, 2021, doi:10.3390/ani11030600_

Round 1

Reviewer 1 Report

A study was designed to investigate the effect of corn and wheat diet in laying hens with 10% dietary Flaxseed and three different multi carbohydrase enzymes on liver inflammation, and alteration in the gut microbiota.

The purpose of the work is clearly stated. The conclusions of the conducted research are clear and

The material used for the research is sufficient, the research methods have been selected appropriately. Table 1 is unreadable, contains duplicate information. The differences between the groups were marked correctly.

Discussing the results against the background of other authors is very detailed.

The publications cited by the authors of the article are well selected. For the most part, the authors refer to the latest knowledge published in renowned scientific journals.

Author Response

Response to the comments of Reviewer 1

Dear Reviewer,

Thank you for your professional opinions and detailed suggestions. We have revised our Manuscript according to the comments. We would like to respond the comments one by one and hope we can make it a better scientific paper and finally published in your great journal.

Thank you again for your careful consideration and excellent suggestions. We hope that our responses are clear and satisfy your requirements.

Yours sincerely,

Professor Xiangmei Zhou

College of Veterinary Medicine,

Key Laboratory of Animal Epidemiology and Zoonosis,

China Agricultural University, Beijing 100193, China

Comments and Suggestions for Authors

A study was designed to investigate the effect of corn and wheat diet in laying hens with 10% dietary Flaxseed and three different multi carbohydrase enzymes on liver inflammation, and alteration in the gut microbiota. The purpose of the work is clearly stated. The conclusions of the conducted research are clear and the material used for the research is sufficient, the research methods have been selected appropriately. Table 1 is unreadable, contains duplicate information. The differences between the groups were marked correctly. Discussing the results against the background of other authors is very detailed. The publications cited by the authors of the article are well selected. For the most part, the authors refer to the latest knowledge published in renowned scientific journals.

Point 1:           Table 1 is unreadable, contains duplicate information.

Response 1:    Thank you very much for your suggestion to improve our manuscript.  Correction has been made as suggested by you. Please see lines 121-122.

Reviewer 2 Report

xx

The manuscript entitled “Effects of corn and wheat diets supplemented with flaxseed and different multi-carbohydrase enzymes on the diversity of cecal microbiota and liver health of laying hens

“represents an important contribution to the understand the relationship between the use of natural feed integration and hen’s production. Intensive hens farming and stock density suppress the immune system, causing loss of product quality, poor growth performance and increasing hen’s mortality rates. So, it is important to tested some natural substance as flaxseed in order to improve the farm management.

The introduction is relevant but it necessary to insert new references. The materials and methods section should be adding more information in order to better the experimental protocol.

The discussion, in the light of the obtained results and of knowledge from is appropriate but it necessary to improve some part.

According to my opinion, the manuscript can be accepted for publication after minor revision.

Below some  corrections are reported.

Title: change it very long title please rewrite it

Abstract: The purpose of the manuscript needs to be improved. Readers must be made to understand that the purpose of the research is to evaluate the effect of corn and diets supplemented with flaxseed on the growth performance and quality of carcass.

Introduction:

Describe the relation between diets supplemented with flaxseed and performances productions in Hens

Insert this reference:

Effect of Adding Flaxseed in the Diet of Laying Hens on Both Production of Omega-3 Enriched Eggs and on Production Performance [2011]

Afaf Y. Al-Nasser; Abdulameer E. Al-Saffar; Faten K. Abdullah; Mariam E. Al-Bahouh; et al

Insert some descriptions on genotype role in microbial composition of laying hens.

See: Chun-Bo Huang et al.,

Rewrite the aim:

No, any study has addressed the association between Flaxseed and enzyme. …. indicate what enzyme  

Materials and Methods

2.4. Experimental design and sample collection

Add information during acclimation (temperature and humidity). During the period of experimental trial was there mortality in hens?

All hens were clinically healthy?

Please described the absence of pathology and clinical parameters.

Enter more details on blood collection: How much time went by before the blood analysis? Indicate storage time ….

How much blood was collected in hens?

How were animals sacrificed? How was the caecal content collected? how is secondary contamination of the microbiota excluded?

Statistics

The statistical analysis used is appropriate

Results:

Reduce it the data showed in tables and figures.

Discussion

The first part is to be included in the introduction, in this section only discussion of the results should be included. Please rewrite the discussions.

Author Response

Response to the comments of Reviewer 2

Dear Reviewer,

Thank you for your professional opinions and detailed suggestions. We have revised our Manuscript according to the comments. We would like to respond the comments one by one and hope we can make it a better scientific paper and finally published in your great journal.

Thank you again for your careful consideration and excellent suggestions. We hope that our responses are clear and satisfy your requirements.

Yours sincerely,

Professor Xiangmei Zhou

College of Veterinary Medicine,

Key Laboratory of Animal Epidemiology and Zoonosis,

China Agricultural University, Beijing 100193, China

Point 1.           The introduction is relevant but it necessary to insert new references. The                                      materials and methods section should be adding more information in order to                                 better the experimental protocol.

Response 1:    Thanks for all valuable suggestions. We hope that these comments will further                              improve the understanding ability of our study. We included new references in the                        introduction section and are highlighted with red color. We also provided the                                 detail procedure for each experiment in the materials and methods section and                             hope that it will be easy for understanding. Please see lines: 48-53, 56-61, 63-72,                                  75-81, 83-87,97-99,102-103,112-127,135-141,160

Point 2.           The discussion, in the light of the obtained results and of knowledge from is appropriate but it necessary to improve some part.

Response 2:    Thanks, we revised the discussion section and properly discuss each result with  the previous findings as suggested by you. Please see lines: 381-391, 393-   404,426-427.

Point 3.           Title: change it very long title please rewrite it.

Response 3:    Title of the manuscript is changed as suggested by you. Please see lines:2-3.

Point 4.           Abstract: The purpose of the manuscript needs to be improved. Readers must be                            made to understand that the purpose of the research is to evaluate the effect of                               corn and diets          supplemented with flaxseed on the growth performance and quality                        of carcass.

Response 4:    Thanks for comments. We revised each section of the manuscript according your                          suggestions.  The main objective and the outcome of the study has been described                         properly. The revised version of the manuscript is clearly addressing the purpose                           and findings of the current study and hence, it will be easy for the reader to                               understand.  Please see lines: 17-18.

Point 5.           Describe the relation between diets supplemented with flaxseed and performances                         productions in Hens

                        Insert this reference:

                        Effect of Adding Flaxseed in the Diet of Laying Hens on Both Production of                                 Omega-3 Enriched Eggs and on Production Performance [2011]. Afaf Y. Al-                                 Nasser; Abdulameer E. Al-Saffar; Faten K. Abdullah; Mariam E. Al-Bahouh; et al

Response 5:    Thanks for suggestion. We properly added the findings of the previous study                                 reported about the relation between diets supplemented with flaxseed and                                      performances productions in Hens in the introduction section. In the revised                                  version of the manuscript the addition of this reference in now mentioned in line                           number 51-53 and the detail reference is on number 501-503 in the reference list.

Point 6.           Insert some descriptions on genotype role in microbial composition of laying                                 hens. See: Chun-Bo Huang et al.,

Response 6:    Thanks for suggestion. We incorporated the study of Chun-Bo Huang et al., in the                         introduction section properly as you suggested. In the revised version of the                                   manuscript the addition of this reference in now mentioned in line number 97-99                           and the detail reference is on number 574-576 in the reference list.

Point 7.           Rewrite the aim: No, any study has addressed the association between Flaxseed                             and enzyme. …. indicate what enzyme.

Response 7:    Thanks for your suggestion. We revised the manuscript and rewrote the major                               objectives and findings clearly. The association of Flaxseed and enzymes are                                 described properly in the results sections. We also mentioned the detail                                          composition of each enzyme in Table 1. Please see lines: 102-103

Point 8.           2.4. Experimental design and sample collection

Add information during acclimation (temperature and humidity). During the period of experimental trial was there mortality in hens?

Response 8:    Thanks for suggestion. Here, in 2.4 section of the materials and methods part we                           describe the design of the experiment. The birds were distributed into various                                experimental groups for different treatment, 15 birds were assigned for each                                  treatment. We didn’t find any mortality during the period of the experimental                                trial. Please see lines: 124-127.

Point 9.           All hens were clinically healthy?

Response 9:    The birds used in the current study were kept for several days before the start of                            the experiment and make sure that all these birds were healthy. Yes, the birds                                were quite healthy at the time of experiment.  

Point 10.         Please described the absence of pathology and clinical parameters.

  Response 10: Thanks for comment. As we selected health bird for our current experiment and kept in a standard environment during the period of the experimental trial. The birds were fed with two diets supplemented with flaxseed and enzymes as described in the materials and methods section. In addition, the birds were protected from harsh environmental conditions and therefore, there was no mortality recorded during the period of experiment. Although, we found inflammatory lesions upon histopathological examination of the liver section, but these lesions were not that much severe to cause mortality. Collectively, the birds were kept in a controlled environment for temperature and humidity according to the standard protocols. In addition, water and feed was provided ad-libitum during the period of experiment. Therefore, we didn’t observe any clear pathological or clinical signs in the birds.   Please see lines:135-141.

Point 11.         Enter more details on blood collection: How much time went by before the blood                          analysis? Indicate storage time ….

Response 11:  Thanks for suggestion. The detail procedure for blood collection, analysis and                               storage time and temperature has been added in the materials and methods section                         of the revised manuscript. The additional information has now mentioned in line                         137-141 numbers of the revised manuscript.

Point 12.         How much blood was collected in hens?

Response 12:  Thanks for your suggestion. Please see line: 137.

Point 13.         How were animals sacrificed? How was the caecal content collected? how is secondary contamination of the microbiota excluded?

Response 13:  Thanks for suggestion. The detail procedure for birds sacrifice, collection of                                  caecal contents and avoiding the contamination of microbiota has been mentioned                         in the materials and methods section. These detail information’s has been                                                 incorporated in the revised manuscript. Please see line: 139

Point 14.         Reduce it the data showed in tables and figures.

Response 14:  Thanks for suggestion. We revised the manuscript and reduced the data shown in tables and in figures as suggested.

Point 15.         The first part is to be included in the introduction, in this section only discussion of the results should be included. Please rewrite the discussions.

Response 15:  Thanks for suggestion. We revised the whole manuscript and rewrote the introduction and discussion section as suggested.

We incorporated all the Changes according to the suggestions. We also reviewed the whole manuscript several times carefully to describe the findings in the text properly and checked spelling, typing and grammatical mistakes. The changes we made in the revised version of manuscript are highlighted with red color. We hope that the revised version of manuscript will be free from any mistakes but if any changes that you might suggest further, we are welcomed.

Reviewer 3 Report

Very interesting study, given the influence of the intestinal microbiota on different aspects of healt and given its dependence on the diet.

The introduction includes the right references, the experimental design is appropriate and the methods and materials used are adequately described, as well as the rules of animal welfare are respected. The results are clearly described and the conclusions supported by the results.

Author Response

Response to the comments of Reviewer 3

Dear Reviewer,

Thank you for your professional opinions and detailed suggestions. We have revised our Manuscript according to the comments. We would like to respond the comments one by one and hope we can make it a better scientific paper and finally published in your great journal.

Thank you again for your careful consideration and excellent suggestions. We hope that our responses are clear and satisfy your requirements.

Yours sincerely,

Professor Xiangmei Zhou

College of Veterinary Medicine,

Key Laboratory of Animal Epidemiology and Zoonosis,

China Agricultural University, Beijing 100193, China

Comments and Suggestions for Authors

Very interesting study, given the influence of the intestinal microbiota on different aspects of health and given its dependence on the diet.

The introduction includes the right references, the experimental design is appropriate, and the methods and materials used are adequately described, as well as the rules of animal welfare are respected. The results are clearly described, and the conclusions supported by the results.

Response: Thank you very much for your comment.

Reviewer 4 Report

This paper has four main criticisms: the harmful effects of flaxseed on gut and liver health and how enzymes could potentially solve such inconvenient is poorly dealt with. The second is that essential parts of the material and methods section are inaccurately described. Thirdly, the way the outcomes are presented does not allow a quick and straightforward interpretation of treatment effects, and the discussion needs to be improved. Fourthly, and not less important, the English language and style must be improved. 

In the introduction, the authors could, and should, improve the description of the consequences of flaxseed utilization in hen nutrition and how enzymes are relevant in such context. This topic has been extensively studied, and literature is, therefore, rich in this regard. The responses assessed by the authors are diverse, meaningful, and interesting for the poultry industry. Nonetheless, the way the topic under study is defended is poor. To improve the quality of the introduction, I suggest the authors follow the sequence described below:

Why use flaxseed in hen diets (enrichment of eggs) > the other side of the story: detrimental effects of such strategy (focus on viscosity) > description with rich details how viscosity impairs digestibility and how undigested nutrients predispose birds to dysbiosis > immune and anti-inflammatory responses when intestinal mucosa recognizes pathogens (flagellin proteins, TLR) > differences in cytokines produced from omega 3 (present in flaxseed) and omega 6 when Coxs hydrolyze membrane phospholipids > deleterious effects of dysbiosis on hen performance > how to solve this problem? (introduce the utilization of enzymes) > highlight how carbohydrases act (remember they hydrolyze different substrates, and it improves digestibility in different ways. Some of them hydrolyze the cell wall, which releases nutrients stuck into cells. In contrast, others will decrease the viscosity, which will allow the access of endogenous enzymes to substrate in the gut lumen and attenuate the physical barrier which impairs the diffusion of nutrients from the intestinal lumen towards the enterocytes > highlight that, even though lots of studies show the advantages of carbohydrases on performance traits and nutrient digestibility, a few publications have addressed the gut microbiome, immune and inflammation biomarkers of hens fed flaxseed based diets > conclude with the objective of the study.

I listed two references below where the authors can find rich information about NSPs on poultry feeds.

Attention to four points!

  1. The authors investigated other enzymes than carbohydrases. Such enzymes do not solve viscosity issues. How could they be fitted into this context? I mean, how could they alleviate the detrimental effects of flaxseed?
  2. Is leptin a pro-inflammatory cytokine? As far as I know, this hormone is produced by adipose tissue that regulates pro-inflammatory cytokines. Careful with some statements.  
  3. Mentioning that no studies were conducted to investigate a given topic is like walking on thin ice! You must be tremendously careful when making such a statement. Several studies correlate NSPs in cereal with enzyme supplementation, including some that investigated specifically carbohydrase supplementation in poultry flaxseed-based diets. Please, find such reviews described below. They must be cited.
  4. Please, note that I am not requesting the authors change the introduction or remove the literature presented. I am just asking for improvements and trying to suggest a better and more logical sequence. Find described below the references mentioned above.

Mathlouthi, N., LalleÌ€s, J.P., Lepercq, P., Juste, C. and Larbier, M., 2002. Xylanase and β-glucanase supplementation improve conjugated bile acid fraction in intestinal contents and increase villus size of small intestine wall in broiler chickens fed a rye-based diet. 

Journal of Animal Science80(11), pp.2773-2779.

Wils-Plotz, E.L., Jenkins, M.C. and Dilger, R.N., 2013. Modulation of the intestinal environment, innate immune response, and barrier function by dietary threonine and purified fiber during a coccidiosis challenge in broiler chicks. Poultry Science92(3), pp.735-745.

Wils-Plotz, E.L. and Dilger, R.N., 2013. Combined dietary effects of supplemental threonine and purified fiber on growth performance and intestinal health of young chicks. Poultry Science92(3), pp.726-734.

Ndou, S.P., Kiarie, E., Thandapilly, S.J., Walsh, M.C., Ames, N. and Nyachoti, C.M., 2017. Flaxseed meal and oat hulls supplementation modulates growth performance, blood lipids, intestinal fermentation, bile acids, and neutral sterols in growing pigs fed corn–soybean meal–based diets. Journal of Animal Science95(7), pp.3068-3078.

Leung, H., Arrazola, A., Torrey, S. and Kiarie, E., 2018. Utilization of soy hulls, oat hulls, and flax meal fiber in adult broiler breeder hens. Poultry science97(4), pp.1368-1372.

Jia, W., Slominski, B.A., Guenter, W., Humphreys, A. and Jones, O., 2008. The effect of enzyme supplementation on egg production parameters and omega-3 fatty acid deposition in laying hens fed flaxseed and canola seed. Poultry science87(10), pp.2005-2014.

Slominski, B.A., Meng, X., Campbell, L.D., Guenter, W. and Jones, O., 2006. The use of enzyme technology for improved energy utilization from full-fat oilseeds. Part II: Flaxseed. Poultry Science85(6), pp.1031-1037.

Yannakopoulos, A.L., Tserveni, G. and Yannakakis, S., 1999. Effect of feeding flaxseed to laying hens on the performance and egg quality and fatty acid composition of egg yolk. Archiv fuer Gefluegelkunde (Germany).

My second criticism is about the material and methods.

Regarding subheading 2.2: this subheading should be named “experimental feeds and enzymes.” In such topic, the authors should describe the feeds and which nutritional levels were considered. This information was included in subheading 2.3. In summary, please, join subheadings 2.2. and 2.3. Start with a description of feeds and then describe the enzymes. The table where the enzymes are described should not contain the group of diets, just the enzymes. Table 2 should include the purity of choline chloride and DL-Met, which is usually 60 and 99%, respectively. Which was the source of L-lysine? Sulfate or HCl? Include the source as well as the purity of it. I also ask for a change in the order the ingredients are listed. Please, do it as follow corn, wheat, soybean meal, flaxseed (meal?), limestone, calcium hydro-phosphate, salt, DL-Methionine, L-Lysine, Mineral premix, vitamin premix, selenium yeast, phytase, antioxidant. Please, include in the footnote the description of the phytase used (FTU).

Regarding the amounts of nutrients, I firstly ask for some changes in the name of nutrients. What is Men Poultry? I imagine this is the metabolizable energy content of diets. If so, the usual abbreviation is AMEn, and the unit Kcal or MJ/kg. Protein should be stated as crude protein. Are the amino acids being expressed in digestible or total base? Please, provide a more accurate description. I wonder if, indeed, these are being expressed as analyzed. It seems that these are being expressed as calculated. I cannot understand why the authors provided the amount of linoleic acid in the composition. Flaxseed is rich in omega-3 fatty acid family, more specifically linolenic acid. Please, inform the content of linolenic acid. In the first raw of the table, there is the name “metric.” Please, replace it with “ingredients.” I would suggest the authors read nutrition papers, and these terminologies would be more familiar.

Regarding subheading 2.4: the authors should describe the experimental design, experimental treatments, and bird husbandry. It could be named “Birds and experimental design.” Please, remove from this topic the sample collection. If I understood well, the treatments were obtained using a 2 x 3 factorial arrangement, where two cereal-based diets were supplemented with three different enzyme blends. If so, inform it similarly to the description I just provided. Please, notify the number of layers used. You could start this subheading as follows: “A total of 540 20-week-old Nongda-3 laying hens were used in a 10-week feeding assay. Six replicate cages of 15 hens were randomly assigned to one of 6 dietary treatments.

Regarding subheading 2.4: this subheading should be named “Sample collection, histological and serum analysis.” In this space, include the information removed from subheading 2.4.

Regarding subheading 2.8: please, improve the description of statistical analysis. I suggest beginning this paragraph as follows: “data were analyzed as two-way ANOVA using the Graph…. When statistical differences among treatment groups were noticed, means were compared using Tukey´s multiple comparison test. Statistical effects were considered when for p<0.05.” I have the feeling that there is something missing in line 135. The description is vague. Please, check this sentence. 

The third, among the four main criticisms highlighted above, is about the way the results are shown.

I imagine the data related to performance responses to experimental diets were/are/will be compiled and exposed in a different paper. There is no problem if this my hypothesis is correct. If this is the case, please, remove body weight data from the current article. Despite, interesting, this is not a response of great interest for egg-laying hens during the egg-laying phase as egg production, egg weight and output, as well as feed conversion is. The treatments were produced as a factorial arrangement, so the effects of diets on responses assessed could be due to cereal type, enzyme supplementation, or the interaction of both. The first sentence in each subheading should be: “There was, or there was not, interactive effects of cereal type and enzyme supplementation on (state the traits assessed).” “Our outcomes indicate that hens fed corn-based diets exhibited a higher liver weight compared with the wheat-based diet-fed group.” These are just examples. Please, follow this pattern for the rest of the result description. In lines 155-157, information is repeated. In figures 2a and 2b, please, fill the bars with colors or textures, which allow a more evident differentiation between bars. They are filled with similar colors.

Another example of poor writing: Line 169-172- “In addition, corn diet vs. wheat diet with the additive of enzyme-c (P < 0.001) showed the highest level of IL-1β followed by enzyme-a (P < 0.01) and enzyme-b (P < 0.05: Figure. 2a), while dietary inclusion of enzyme-a and b in corn diet led to highest expression (P < 0.001) of IL-6.” I cannot understand what the authors said. Please, be clear about interactive effects and describe the interactions. I will not continue with this line by line review. Please, use these hints to improve the rest of the result section.

Discussion

In line 359, the authors should reconsider this statement, once they did not investigate carbohydrase complexes exclusively. The enzymatic complex b, for example, contains just proteases (neutral and alkaline).

Line 361-363: Such a statement cannot be accepted. When diets are balanced for nutrients, mainly energy: lysine ratio, there is no “fattener diet” or “fattener ingredient.” The energy produced from the oxidation of organic compounds is used for maintenance, body protein accretion, and, in laying hens, for egg protein synthesis. If the amounts of nutrients are balanced, energy will be used for such functions, regardless of the cereal used. Differences in body composition are generally explained by high energy contents and inadequate amino acid supply. The findings obtained in the current assay, that corn diets elicited an increase in fat deposition compared with wheat diets may be due to differences in the balance between energy and protein, i.e., amino acids, which might be associated with differences in nutrient balance or differences in the efficiency of nutrient utilization. Are the authors sure that both diets are isonutritive? If so, one possible explanation for the differences in fat deposition is that energy in corn diets is more efficiently used than that in wheat diets due to the lower viscosity of corn diets. Presumably, such diets were formulated with more energy than required, so after meeting maintenance, growth, and egg production requirements and losing part of energy and heat, the extra amounts were stored as bodily lipids. In wheat diets, the digestibility of nutrients, mainly lipids, is impaired by viscosity. Consequently, the amounts of energy available to be stored as body lipids may be lower than those in corn diets. In summary, there are no fattener ingredients, but rather imbalances in nutrient composition and differences in the digestion and absorption of such nutrients, which may alter the final amounts of net energy available for maintenance and production purposes.

Please, remove the mention that corn feeds elicit fat accumulation in the liver. It is not valid. In almost the entire American continent, for example, practical feeds are formulated based on corn. The reasons underlying fat accumulation in the liver are several and include, for example, mycotoxin exposure, deficiency of methyl group donors such as choline and methionine, etc. What was the ether extract, i.e., fat content, of the diets? Such differences may also have some correlation to dietary fat content.

Line 363 – 366: Please, improve the discussion with good references and write more about how corn diets triggered the production of inflammatory cytokines compared to wheat. Usually, the opposite has proven right since wheat-based feeds predispose chicks to dysbiosis due to high viscosity.

Line 366 – 370: the statements are vague. Please, which cereals and enzymes are the authors talking about? I would advise focusing on the enzymes and grain used in your assay. Even though discrepancies in the literature exist, there are trustworthy and reliable outcomes involving cereals and enzymes, and usually, the effects are not so diverse.

Line 370: flaxseed main antinutritional factor is mucilage, which, as well as arabinoxylans, increases the viscosity of digesta. The authors should mention it.

Line 370 – 375: I would not say that the absence of deleterious effects of flaxseed could be attributed to enzyme efficiency. At 15% of inclusion, flaxseed has proved to be safe to ensure proper poultry performance. The authors can state it only if they have tested all the diets without enzyme supplementation. Otherwise, statements and conclusions can only be made for cereal type and enzyme type.

Line 396 – 397: how can the readers reach such a conclusion if the authors did not provide other performance response data than body weight, which, by the way, as highlighted before, is not a response of great concern for the egg-laying cycle phase.

In the rest of the discussion, the authors do not associate causes and effects but rather repeat the outcomes. How the cereals and enzymes used in diet formulation affect the gut microbiome? Which findings in the literature support yours? Why other enzymes than carbohydrases were used alone? Why improve the digestion of protein? The authors could have explored that undigested amino acids such as glycine and methionine may be fermented in the distal ileum and hindgut by pathogens such as clostridium perfringens and Escherichia coli. It could have been mentioned in the introduction and better explored in the discussion session.

Finally, English needs to be improved when it comes to grammar, spelling, punctuation, consistency, and formality. The readability of the paper should be improved. 

Author Response

Response to Reviewer 4

Dear Reviewer,

Thank you for your professional opinions and detailed suggestions. We have revised our Manuscript according to the comments. We would like to respond the comments one by one and hope we can make it a better scientific paper and finally published in your great journal.

Thank you again for your careful consideration and excellent suggestions. We hope that our responses are clear and satisfy your requirements.

Yours sincerely,

Professor Xiangmei Zhou

College of Veterinary Medicine,

Key Laboratory of Animal Epidemiology and Zoonosis,

China Agricultural University, Beijing 100193, China

Comments and Suggestions for Authors

Point: 1: Nonetheless, the way the topic under study is defended is poor. To improve the quality of the introduction, I suggest the authors follow the sequence described below:

Introduction

Why use flaxseed in hen diets (enrichment of eggs) > the other side of the story: detrimental effects of such strategy (focus on viscosity). > description with rich details how viscosity impairs digestibility and how undigested nutrients predispose birds to dysbiosis > immune and anti-inflammatory responses when intestinal mucosa recognizes pathogens (flagellin proteins, TLR)> differences in cytokines produced from omega 3 (present in flaxseed) and omega 6 when Coxs hydrolyze membrane phospholipids > deleterious effects of dysbiosis on hen performance > how to solve this problem? (introduce the utilization of enzymes) > highlight how carbohydrases act (remember they hydrolyze different substrates, and it improves digestibility in different ways. Some of them hydrolyze the cell wall, which releases nutrients stuck into cells. In contrast, others will decrease the viscosity, which will allow the access of endogenous enzymes to substrate in the gut lumen and attenuate the physical barrier which impairs the diffusion of nutrients from the intestinal lumen towards the enterocytes > highlight that, even though lots of studies show the advantages of carbohydrases on performance traits and nutrient digestibility, a few publications have addressed the gut microbiome, immune and inflammation biomarkers of hens fed flaxseed based diets > conclude with the objective of the study.

I listed two references below where the authors can find rich information about NSPs on poultry feeds.

Please, note that I am not requesting the authors change the introduction or remove the literature presented. I am just asking for improvements and trying to suggest a better and more logical sequence. Find described below the references mentioned above.

Mathlouthi, N., LalleÌ€s, J.P., Lepercq, P., Juste, C. and Larbier, M., 2002. Xylanase and β-glucanase supplementation improve conjugated bile acid fraction in intestinal contents and increase villus size of small intestine wall in broiler chickens fed a rye-based diet. 

Journal of Animal Science80(11), pp.2773-2779.

Yannakopoulos, A.L., A.S. Tserveni-Gousi and S. Yannakakis, 1999. Effect of feeding flaxseed to laying hens on the performance and egg quality and fatty acid composition of egg yolk. Archives fur Geflügelkunde, 63: 260-263.

Wils-Plotz, E.L., Jenkins, M.C. and Dilger, R.N., 2013. Modulation of the intestinal environment, innate immune response, and barrier function by dietary threonine and purified fiber during a coccidiosis challenge in broiler chicks. Poultry Science92(3), pp.735-745.

Wils-Plotz, E.L. and Dilger, R.N., 2013. Combined dietary effects of supplemental threonine and purified fiber on growth performance and intestinal health of young chicks. Poultry Science92(3), pp.726-734.

Ndou, S.P., Kiarie, E., Thandapilly, S.J., Walsh, M.C., Ames, N. and Nyachoti, C.M., 2017. Flaxseed meal and oat hulls supplementation modulates growth performance, blood lipids, intestinal fermentation, bile acids, and neutral sterols in growing pigs fed corn–soybean meal–based diets. Journal of Animal Science95(7), pp.3068-3078.

Leung, H., Arrazola, A., Torrey, S. and Kiarie, E., 2018. Utilization of soy hulls, oat hulls, and flax meal fiber in adult broiler breeder hens. Poultry science97(4), pp.1368-1372.

Jia, W., Slominski, B.A., Guenter, W., Humphreys, A. and Jones, O., 2008. The effect of enzyme supplementation on egg production parameters and omega-3 fatty acid deposition in laying hens fed flaxseed and canola seed. Poultry science87(10), pp.2005-2014.

Slominski, B.A., Meng, X., Campbell, L.D., Guenter, W. and Jones, O., 2006. The use of enzyme technology for improved energy utilization from full-fat oilseeds. Part II: Flaxseed. Poultry Science85(6), pp.1031-1037.

Response 1:    Thank you for your detailed suggestion. We have revised manuscript by adding                             references suggested by you. Please see lines: 48-53, 56-61, 63-72,                                                 75-81, 83-87,97-99,102-103.

Attention to four points!

Point 2:  The authors investigated other enzymes than carbohydrases. Such enzymes do not                        solve viscosity issues. How could they be fitted into this context? I mean, how could                        they alleviate the detrimental effects of flaxseed?

Response 2: Thank you very much for your concern. We have revised our manuscript                                          as suggested by you.

Point 3:           Is leptin a pro-inflammatory cytokine? As far as I know, this hormone is produced                        by adipose tissue that regulates pro-inflammatory cytokines. Careful with some                             statements.  

Response 3: Thank you very much for pointing our mistake. We have revised introduction                                   as suggested by you.

Point 4:           Mentioning that no studies were conducted to investigate a given topic is like                                walking on thin ice! You must be tremendously careful when making such a                                  statement. Several studies correlate NSPs in cereal with enzyme supplementation,                                     including some that investigated specifically carbohydrase supplementation in                               poultry flaxseed-based diets. Please, find such reviews described below. They                               must be cited.

Response 4:    Thank you very much for pointing our mistake. We have cited papers in the                                   introduction as suggested by you.

Point 5:           Please, note that I am not requesting the authors change the introduction or                                    remove the literature presented. I am just asking for improvements and trying to                            suggest a better and more logical sequence. Find described below the references                            mentioned above.

Response 5: Thank you very much for your valuable suggestions. We have cited papers as                                   suggested by you.

My second criticism is about the material and methods.

Point 6:           Regarding subheading 2.2: this subheading should be named “experimental feeds                          and enzymes.” In such topic, the authors should describe the feeds and which                                nutritional levels were considered. This information was included in subheading                            2.3. In summary, please, join subheadings 2.2. and 2.3. Start with a description of                                feeds and then describe the enzymes.

Response 6:    Thank you very much for suggestion. We have merged subheading 2.2 and 2.3                              into one table. Please see lines: 112-122.

Point 7.           Table 2 should include the purity of choline chloride and DL-Met, which is                                   usually 60 and 99%, respectively. Which was the source of L-lysine? Sulfate or                             HCl? Include the source as well as the purity of it. Which was the source of L-                            lysine? Sulfate or HCl

Response 7:    Thank you very much for suggestion. We have revised manuscript as suggested                            by you. The source of L-lysine is sulfate.

Point 8:           I also ask for a change in the order the ingredients are listed. Please, do it as                                   follow corn, wheat, soybean meal, flaxseed (meal?), limestone, calcium hydro-                             phosphate, salt, DL-Methionine, L-Lysine, Mineral premix, vitamin premix,                                  selenium yeast, phytase, antioxidant. Please, include in the footnote the                                          description of the phytase used (FTU).

Response 8:    Thank you very much for suggestion. We have changed in the order of the                                     ingredients listed. Please see lines: 121-122.

Point 9:           Regarding the amounts of nutrients, I firstly ask for some changes in the name of                         nutrients. What is Men Poultry? I imagine this is the metabolizable energy content                        of diets. If so, the usual abbreviation is AMEn, and the unit Kcal or MJ/kg.

Response 9:    Thank you very much for suggestion. Men Poultry is a nutrient and unit is mc/kg. 

Point 10:         Protein should be stated as crude protein. Are the amino acids being expressed in                          digestible or total base? Please, provide a more accurate description. I wonder if,                           indeed, these are being expressed as analyzed. It seems that these are being                                    expressed as calculated.

Response 10: Thank you very much for your suggestion. We have revised our manuscript as                               suggested by you. Please see lines: 121-122.

Point 11:         I cannot understand why the authors provided the amount of linoleic acid in the                             composition. Flaxseed is rich in omega-3 fatty acid family, more specifically                                 linolenic acid. Please, inform the content of linolenic acid.

Response 11: Thank you very much for your concern. Thank you very much for your                                          suggestion. Yes, flaxseed is rich in omega-3 fatty acid family, more specifically                            linolenic acid. Its contents are linoleic acid and other DHA, EPA.

Point 12:         In the first raw of the table, there is the name “metric.” Please, replace it with                                “ingredients.” I would suggest the authors read nutrition papers, and these                                      terminologies would be more familiar.

Response 12: Thank you very much for your suggestion. We have replaced metric with                                       ingredients as suggested by you. Please see lines: Please see lines: 121-122.

Point 13:         Regarding subheading 2.4: the authors should describe the experimental design,                            experimental treatments, and bird husbandry. It could be named “Birds and                                    experimental design.” Please, remove from this topic the sample collection. If I                             understood well, the treatments were obtained using a 2 x 3 factorial arrangement,                        where two cereal-based diets were supplemented with three different enzyme                                blends. If so, inform it similarly to the description I just provided. Please, notify                            the number of layers used. You could start this subheading as follows: “A total of                               540 20-week-old Nongda-3 laying hens were used in a 10-week feeding assay.                              Six replicate cages of 15 hens were randomly assigned to one of 6 dietary                                             treatments.

Response 13: Thank you very much for your suggestion. We have revised our manuscript as                               suggested by you. Please see lines: 123-127.

Point 14:         Regarding subheading 2.4: this subheading should be named “Sample collection,                          histological and serum analysis.” In this space, include the information removed                            from subheading 2.4.

Response 14: Thank you very much for your suggestion. We have renamed subheadings as                                 suggested by you. Please see lines: 135-141.

Point 15:         Regarding subheading 2.8: please, improve the description of statistical analysis. I                        suggest beginning this paragraph as follows: “data were analyzed as two-way                                ANOVA using the Graph…. When statistical differences among treatment groups                         were noticed, means were compared using Tukey´s multiple comparison test.                             Statistical effects were considered when for p<0.05.” I have the feeling that there                               is something missing in line 135. The description is vague. Please, check this                              sentence.

Response 15: Thank you very much for your suggestion. We have rewritten statistical analysis.                          Please see line: 160.

 Point 16:        The third, among the four main criticisms highlighted above, is about the way the                         results are shown. I imagine the data related to performance responses to                                        experimental diets were/are/will be compiled and exposed in a different paper.                              There is no problem if this my hypothesis is correct. If this is the case, please,                                remove body weight data from the current article.

Response 16:  Thank you very much for your suggestion. We have removed body weight data                            from manuscript.

Point 17:         Despite, interesting, this is not a response of great interest for egg-laying hens                               during the egg-laying phase as egg production, egg weight and output, as well as                           feed conversion is. The treatments were produced as a factorial arrangement, so                                    the effects of diets on responses assessed could be due to cereal type, enzyme                              supplementation, or the interaction of both. The first sentence in each subheading                            should be: “There was, or there was not, interactive effects of cereal type and                               enzyme supplementation on (state the traits assessed).” “Our outcomes indicate                           that hens fed corn-based diets exhibited a higher liver weight compared with the                              wheat-based diet-fed group.” These are just examples. Please, follow this pattern                               for the rest of the result description.

Response 17: Thank you very much for your suggestion. We have revised our manuscript as                               suggested by you.

Point 18:         In lines 155-157, information is repeated. In figures 2a and 2b, please, fill the bars                        with colors or textures, which allow a more evident differentiation between bars.                           They are filled with similar colors.

Response 18: Thank you very much for your suggestion. We have revised our manuscript as                               suggested by you.

Point 19:         Another example of poor writing: Line 169-172- “In addition, corn diet vs. wheat                          diet with the additive of enzyme-c (P < 0.001) showed the highest level of IL-1β                           followed by enzyme-a (P < 0.01) and enzyme-b (P < 0.05: Figure. 2a), while                                 dietary inclusion of enzyme-a and b in corn diet led to highest expression (P <                               0.001) of IL-6.” I cannot understand what the authors said. Please, be clear about                          interactive effects and describe the interactions. I will not continue with this line                           by line review. Please, use these hints to improve the rest of the result section.

Response 19: Thank you very much for your suggestion. We have revised our manuscript as                               suggested by you. Please see lines: 185-188.

Discussion

Point 20:         In line 359, the authors should reconsider this statement, once they did not                                     investigate carbohydrase complexes exclusively. The enzymatic complex b, for                             example, contains just proteases (neutral and alkaline).

Response 20: Thank you very much for your suggestion. We have revised whole manuscript.

Point 21:         Line 361-363: Such a statement cannot be accepted. When diets are balanced for                           nutrients, mainly energy: lysine ratio, there is no “fattener diet” or “fattener                                   ingredient.” The energy produced from the oxidation of organic compounds is                               used for maintenance, body protein accretion, and, in laying hens, for egg protein                                  synthesis. If the amounts of nutrients are balanced, energy will be used for such                                     functions, regardless of the cereal used. Differences in body composition are                               generally explained by high energy contents and inadequate amino acid supply.                          The findings obtained in the current assay, that corn diets elicited an increase in                                 fat deposition compared with wheat diets may be due to differences in the balance                   between energy and protein, i.e., amino acids, which might be associated with                             differences in nutrient balance or differences in the efficiency of nutrient                                        utilization.

Response 21: Thank you very much for your suggestion. We have revised our manuscript as                               suggested by you. Please see lines: 391-392.

Point 22:         Are the authors sure that both diets are isonutritive? If so, one possible                                           explanation for the differences in fat deposition is that energy in corn diets is                                 more efficiently used than that in wheat diets due to the lower viscosity of corn                           diets. Presumably, such diets were formulated with more energy than required, so                                after meeting maintenance, growth, and egg production requirements and losing                                    part of energy and heat, the extra amounts were stored as bodily lipids. In wheat                                  diets, the digestibility of nutrients, mainly lipids, is impaired by viscosity.                                              Consequently, the amounts of energy available to be stored as body lipids may be                                  lower than those in corn diets. In summary, there are no fattener ingredients, but                                  rather imbalances in nutrient composition and differences in the digestion and                           absorption of such nutrients, which may alter the final amounts of net energy                              available for maintenance and production purposes. Please, remove the mention                            that corn feeds elicit fat accumulation in the liver. It is not valid. In almost the                               entire American continent, for example, practical feeds are formulated based on                            corn. The reasons underlying fat accumulation in the liver are several and include,                      for example, mycotoxin exposure, deficiency of methyl group donors such as                             choline and methionine, etc. What was the ether extract, i.e., fat content, of the                            diets? Such differences may also have some correlation to dietary fat content.

Response 22: Thank you very much for your suggestion. We have removed it as suggested by                             you.

Point 23:         Line 363 – 366: Please, improve the discussion with good references and write                             more about how corn diets triggered the production of inflammatory cytokines                              compared to wheat. Usually, the opposite has proven right since wheat-based                                feeds predispose chicks to dysbiosis due to high viscosity.

Response 23: Thank you very much for your suggestion. We have improved discussion as                                   suggested by you. Please see lines: 381-404, 425-426.

Point 24:         Line 366 – 370: the statements are vague. Please, which cereals and enzymes are                           the authors talking about? I would advise focusing on the enzymes and grain used                                     in your assay. Even though discrepancies in the literature exist, there are                                        trustworthy and reliable outcomes involving cereals and enzymes, and usually, the                        effects are not so diverse.

Response 24: Thank you very much for your suggestion. We have revised our manuscript as                               suggested by you. Please see lines: 381-404, 425-426.

Point 25:         Line 370: flaxseed main antinutritional factor is mucilage, which, as well as                                   arabinoxylans, increases the viscosity of digesta. The authors should mention it.

Response 25: Thank you very much for your suggestion. We have revised our manuscript as                               suggested by you. Please see lines: 381-404, 425-426.

Point 26:         Line 370 – 375: I would not say that the absence of deleterious effects of flaxseed                                     could be attributed to enzyme efficiency. At 15% of inclusion, flaxseed has                                   proved to be safe to ensure proper poultry performance. The authors can state it                             only if they have tested all the diets without enzyme supplementation. Otherwise,                                  statements and conclusions can only be made for cereal type and enzyme type.

Response 26: Thank you very much for your suggestion. We have revised our manuscript as                               suggested by you. Please see lines: 381-404, 425-426.

Point 27:         Line 396 – 397: how can the readers reach such a conclusion if the authors did not                        provide other performance response data than body weight, which, by the way, as                         highlighted before, is not a response of great concern for the egg-laying cycle                                phase. In the rest of the discussion, the authors do not associate causes and effects                         but rather repeat the outcomes. How the cereals and enzymes used in diet                                             formulation affect the gut microbiome? Which findings in the literature support                                     yours? Why other enzymes than carbohydrases were used alone? Why improve                                   the digestion of protein? The authors could have explored that undigested amino                               acids such as glycine and methionine may be fermented in the distal ileum and                                    hindgut by pathogens such as clostridium perfringens and Escherichia coli. It                                 could have been mentioned in the introduction and better explored in the                                               discussion session.

Response 27: Thank you very much for your suggestion. We have revised as suggested by you.                          Please see lines: 381-404, 425-426.

Point 28:         Finally, English needs to be improved when it comes to grammar, spelling,                                    punctuation, consistency, and formality. The readability of the paper should be                              improved.

Response 28: Thank you very much for your suggestion. We have revised our manuscript as                               suggested by you.